# ProteinConformers: Benchmark Dataset for Simulating Protein Conformational Landscape Diversity and Plausibility

**Yihang Zhou**[1*]   **Chen Wei**[1,2*]   **Minghao Sun**[1]   **Jin Song**[3,4]   **Yang Li**[1]
**Lin Wang**[5†]   **Yang Zhang**[1,5†]

[1]National University of Singapore   [2]Xi'an University of Posts & Telecommunications
[3]University of Chinese Academy of Sciences   [4]Chinese Academy of Sciences
[5]Chinese Academy of Medical Sciences

yihangjoe@foxmail.com, weichen@xupt.edu.cn
wenzhao30030@gmail.com, zhang@nus.edu.sg

## Abstract

Understanding the conformational landscape of proteins is essential for elucidating protein function and facilitating drug design. However, existing protein conformation benchmarks fail to capture the full energy landscape, limiting their ability to evaluate the diversity and physical plausibility of AI-generated structures. We introduce **ProteinConformers**, a large-scale benchmark dataset comprising over **381,000** physically realistic conformations for **87** CASP targets. These were derived from more than **40,000** structural decoys via extensive all-atom molecular dynamics simulations totaling over **6 million CPU hours**. Using this dataset, we propose novel metrics to evaluate conformational diversity and plausibility, and systematically benchmark six protein conformation generative models. Our results highlight that leveraging large-scale protein sequence data can enhance a model's ability to explore conformational space, potentially reducing reliance on MD-derived data. Additionally, we find that PDB and MD datasets influence model performance differently, current models perform well on inter-atomic distance prediction but struggle with inter-residue orientation generation. Overall, our dataset, evaluation metrics, and benchmarking results provide the first comprehensive foundation for assessing generative models in protein conformational modeling. Dataset and instructions are available at `https://huggingface.co/datasets/Jim990908/ProteinConformers/tree/main`. Codes are stored at `https://github.com/auroua/ProteinConformers`. An interactive website locates at `https://zhanggroup.org/ProteinConformers`.

## 1   Introduction

In recent years, the field of protein structure prediction has rapidly shifted from a "single-conformation" paradigm toward "multi-conformation" modeling. While energy-based and deep learning–based methods such as I-TASSER[1], AlphaFold[2] and RoseTTAFold[3] deliver accurate static 3D structures, proteins under physiological conditions typically sample multiple functionally relevant states. The prediction of protein multi-conformations from conformational landscape is critical both for understanding molecular mechanisms and for demanding downstream applications. Yet, most studies to date have paid little systematic attention to balancing a model's ability to generate multiple conformers with conformational diversity, or to examine the atom-level physical plausibility.

---

[*]Equal contribution.   [†] Corresponding authors.

39th Conference on Neural Information Processing Systems (NeurIPS 2025) Track on Datasets and Benchmarks.

There remains no benchmark dataset or metric suite that faithfully emulates a realistic folding funnel while simultaneously capturing polymorphism and ensuring physical plausibility.

To date, there is no analogous benchmark for multi-conformation generation. For single-conformation prediction, the Protein Data Bank (PDB)[4] serves as the golden standard, and its experimentally determined "native" structures naturally occupy global minima in the folding funnel. Some efforts have attempted to harvest a conformational ensemble by running extensive all-atom molecular dynamics (MD) simulations from the native state, yielding a wealth of near-native fluctuations around the crystallographic structure. However, the energy-based nature of MD inherently biases sampling toward the initial funnel basin[5, 6], and only extremely long simulations have any chance of escaping into more remote minima—at a computational cost that is both prohibitive and difficult to predict.

In terms of evaluation metrics, the community has established measures to assess single-conformation quality, such as TM-score[7], RMSD[8], and LDDT[9]. Metrics tailored to multi-conformation ensembles, however, remain in infancy. Recent proposals compute Jensen-Shannon (JS) divergence between ensembles based on pairwise distance distributions, or radius of gyration histograms, but still fall short of capturing atom-level distance and angle plausibility. Since a protein's conformation is uniquely defined by its internal geometry, a truly comprehensive metrics should directly evaluate the completeness and physical realism of all interatomic distance and torsional angles.

To address these gaps, we introduce **ProteinConformers** (Figure1). Starting from 87 Critical Assessment of protein Structure Prediction (CASP) [10, 11, 12] targets, we collected and filtered over 40,000 decoy conformers generated by hundreds of different traditional and AI based prediction algorithms. Each decoy was independently refined with all-atom MD simulations to resolve steric clashes and fine-tune geometry, yielding more than 381,000 physically realistic conformations that both span diverse regions of the folding landscape and satisfy stringent physicochemical constraints. To our knowledge, this is the first dataset of its kind, and none of the 87 proteins we release appear in prior multi-conformer benchmarks. Building this resource consumed over **6 million CPU hours**.

Built on this diverse and trustful dataset, we design a dual-axis evaluation framework. Axis one is **Conformational Diversity**, to quantify the breadth and representativeness of generated conformers across the overall energy funnel, with corresponding metrics. The axis is **Conformational Physics Plausibility**, which assesses the all-atom geometric and energetic validity of every generated conformer, with dedicated metrics.

Our preliminary benchmarking of six approaches[13, 14, 15] reveals that, while many methods accurately reproduce interatomic distance distributions, they generally under-explore the broader conformational landscape and exhibit significant shortcomings in torsion-angle sampling. Besides the basic task of ProteinConformers to understand how well current conformational generative methods explore the broader conformational landscape, this work seeks to provide some insight to the question: *Is it possible to develop generative models capable of superior energy landscape exploration without reliance on molecular dynamics (MD) simulation datasets?* By releasing **ProteinConformers** and its accompanying evaluation framework, we provide a large-scale, high-fidelity benchmark that will accelerate advances in protein structure prediction, multi-conformation modeling, computational biology, and rational drug discovery.

## 2 Related Work

### 2.1 Protein Conformational Landscape Exploration

Early efforts to map the protein folding funnel relied on coarse-grained sampling methods that rotate backbone or side-chain dihedrals under simplified force fields [16, 17], as well as fragment-assembly threading guided by Monte Carlo Simulation[18] algorithm, such as 3DRobot [19]. Although computationally efficient, these approaches suffer from low-accuracy energy functions and heavily reduced representations of protein geometry. To improve physical fidelity, recent work such as AIMD-Chig[20] has explored conformational space at the DFT level [21], but only for a single 166-atom peptide due to the prohibitive cost. Likewise, large-scale MD simulations initialized from experimental structures, such as BPTI [22], Atlas [23], mdCATH [24], Dynamic PDB [25], provide ensembles of near-native fluctuations, yet these trajectories remain confined to the primary funnel basin unless run for impractically long time, limiting exploration of more diverse conformations. Unlike previous studies, we run MD simulations in batch with decoys at variant positions in energy

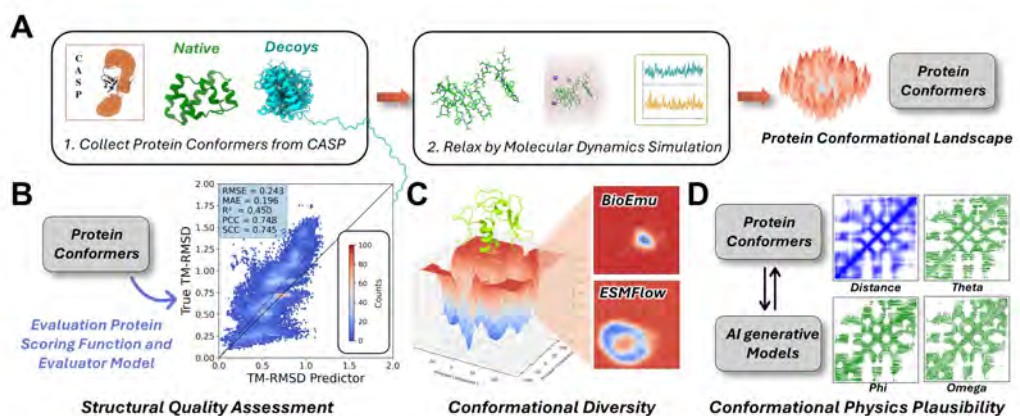

Figure 1: **Overview of the ProteinConformers benchmark and evaluation framework**. (A) We curate decoy ensembles for CASP targets and, for each decoy, perform all-atom MD refinement to sample across the folding landscape. (B) Beyond providing conformations, ProteinConformers includes a panel of per-conformation energy metrics, enabling auxiliary assessment of decoy quality. (C) We define diversity metrics that quantify how well generated conformations explore the overall conformational funnel. (D) We design physics-plausibility metrics to evaluate the atom-level realism based on inter-residual distances and torsion angles.

landscapes generated by hundreds of different algorithms, which guarantees both diversity and plausibility of protein conformational landscape exploration.

## 2.2 Protein Conformation Generative Models

Traditional generative strategies commonly adopt coarse-grained movements to synthesize conformers [26, 27], but such models frequently produce steric clashes, and require post hoc side-chain reconstruction. With the advent of highly accurate single-structure predictors, a new class of diffusion [28, 29] and flow-based [30] generative models has emerged, fine-tuning backbones or post-processing predicted poses to yield all-atom ensembles. Examples include AlphaFlow [13], ESMFlow [13], BioEmu [14] and ESMDiff [15]. AlpahFlow and ESMFlow are two protein conformation generative models obtained by fine-tuning AlphaFold2 [2] and ESMFold [31] using flow matching framework, respectively. BioEmu [14] uses the trained AlphFold2 model to extract features and using a two-step training strategy to train a denoise diffusion model based on the extracted features to generate a collection of conformations that can reflect the equilibrium distribution of the structure of input sequences. ESMDiff [15] is fine-tuned from pre-trained ESM3 [32], and uses a conditional language model to capture sequence-specific conformational distributions. AFsample2 [33] generates ensembles by MSA (multiple sequence alignment) sampling based on AlphaFold2. AlphaFold3 [34] produces multiple conformations by initiating random noises during diffusion inference.

# 3 ProteinConformers Dataset

## 3.1 Source of ProteinConformers

CASP is a biennial competition and the CASP committee curates dozens of representative targets and invite global teams to predict the structures under blind conditions. The resulting predictions reflect the cutting-edge-tech in protein structure prediction and exhibit high conformational diversity due to the variety of methods employed by participating teams, which collectively enables extensive sampling of the protein folding landscape. However, these predicted structures are not guaranteed to be physically realistic at the atomic level. The MD simulation, by contrast, use physics-based force fields to refine protein conformations at full atomic resolution. Nevertheless, MD simulations are computationally expensive and conformational transitions are often limited to regions near the input structure or trapped in local minima[5, 6], resulting in high redundant conformations.

To address these complementary limitations, by integrating the strengths of both resources, we compiled the predictions from CASP and, after quality control, curated a dataset of 40,387 predicted structures for 87 proteins, and then performed MD simulations on each structure. Taking advantage of the large conformational shifts observed in the early phase of MD trajectories, we limited simulations to short timescales to reduce computational cost and redundancy. This resulted in a final dataset of 381,546 all-atom, physically refined protein conformations broadly sampling the folding landscape.

## 3.2 Preprocessing of Protein Conformers

To build ProteinConformers benchmark, we systematically collected, cleaned and corrected all prediction entries in CASP14 [11] and CASP15 [12]. In CASP14, nearly 100 teams participated in predicting structures for 90 modeling targets. CASP15 had a similar number of participants working on 127 targets. The predicted protein structures are processed according to the following steps.

**First**, we remove duplicate accessions, non-protein entries and redundant sequences to obtain 172 unique targets with corresponding predictions. **Second**, for each prediction, chain sequences are extracted with Biopython [35] and aligned to the corresponding reference. The alignment yields three mutually exclusive categories: *same* means full-length and residue-order identity; *disorder* means a contiguous subsequence of the reference (internal or terminal deletions, order preserved); *mismatch* means any substitution, insertion or reordered segment. Oligomeric or hetero-complex models are annotated and excluded from the benchmark. **Third**, based on the categories from the previous step, the *same* models are accepted without modification. The *disorder* models are retained only if their residue numbers already matched the reference structure; otherwise the coordinates are automatically renumbered to restore one-to-one correspondence. All *mismatch* entries are discarded. **Fourth**, when multiple experimental native structures exists, pairwise TM-scores are calculated to chose the reference native. Targets without a definitive match are resolved by manual inspection. **Fifth**, a target enters the benchmark only if it had at least one native structure and 100 decoys. For targets below this threshold, additional decoys are generated using 3DRobot [19].

## 3.3 MD Simulation Protocol

MD simulations are performed using GROMACS 2023 [36]. Each protein conformer follows the same workflow. **Topology construction and solvation:** The OPLS–AA force field is used for topology generation, together with the TIP3P water model. Each protein is centered in a dodecahedral box, and the box is filled with pre-equilibrated SPC216 water. $Na^+$ and $Cl^-$ ions are added to neutralize total charge. **Energy minimization:** Steepest-descent minimization is applied until the largest force on any atom fell below $1000\,\mathrm{kJ\,mol^{-1}\,nm^{-1}}$ (maximum 50,000 steps), thereby eliminating steric clashes and unrealistic geometries. **Two-stage equilibration:** The NVT phase ($100\,\mathrm{ps},\ 300\,\mathrm{K}$) employed the V-rescale thermostat with positional restraints on heavy atoms to stabilize temperature. The subsequent NPT phase ($100\,\mathrm{ps},\ 1\,\mathrm{bar}$) uses the Parrinello–Rahman barostat after partially releasing restraints to equilibrate density. **Run and sampling:** Restraints are removed and a simulation between 125 ps to 375 ps is executed. Periodic-boundary artifacts are eliminated by recentering and re-imaging the trajectory. Snapshots are extracted every 25 ps. System energies (total, potential, and protein-only) are recorded throughout for subsequent energy-landscape modeling. **Computational resources:** Simulations are executed on high-performance computers equipped with AMD EPYC 7763 (64-core, 2.45 GHz) processors. Applying this protocol to 40,387 starting conformers from 87 proteins generated 381,546 all-atom refined conformations, at an aggregate cost over 6 million CPU hours. These trajectories and associated physical properties constitute the core of the ProteinConformers.

## 3.4 Analysis of ProteinConformers Dataset

ProteinConformers benchmark dataset is challenging. It comprises 87 CASP targets with an average sequence length of 305 residues and a median of 255 (Figure 2A). These targets, selected by the CASP committee, span a broad array of fold topologies—including predominantly α-helical proteins, β-sheet–rich folds, disordered regions, and multi-domains. Notably, 32 of the targets exceed 300 residues, and the largest reaches 949 residues. ProteinConformers provides both diverse and balanced protein conformational landscape ensembles. Each protein sampled 4,386 conformations on average. By classifying conformers as near-native (TM-score $\geq 0.5$) or non-native (TM-score $< 0.5$), we show that the dataset spans the full spectrum of conformational landscape, as visualized by

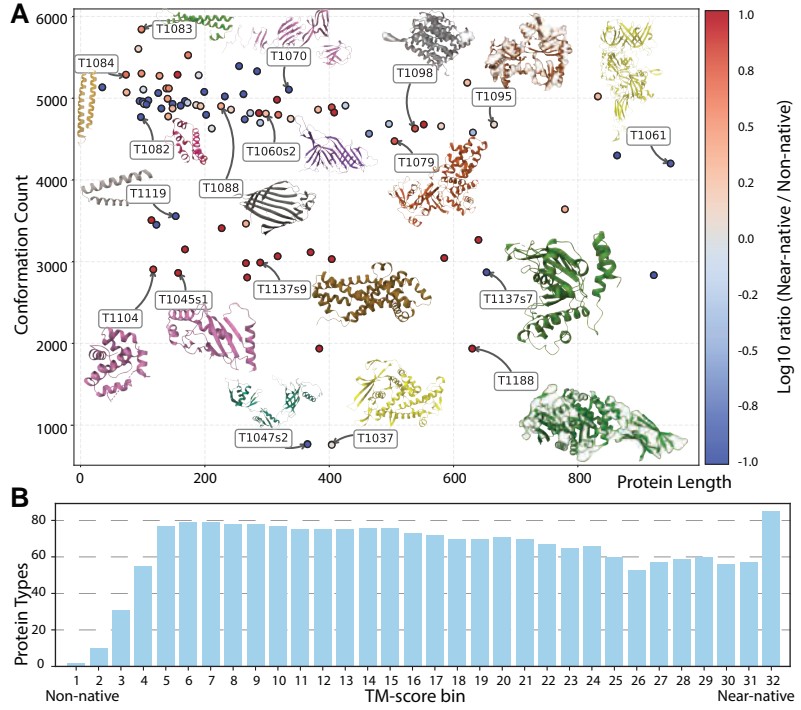

Figure 2: **Global distribution of the ProteinConformers dataset**. (A) Number of conformers per protein, sorted by sequence length (x-axis). The y-axis shows the total distinct conformational states captured for each target. Bar color encodes the log10-ratio of near-native to non-native conformations in our benchmark. Insets display 3D renderings of randomly selected proteins. (B) TM-score coverage histogram across the dataset. The x-axis divides TM-score into 32 equal-width bins from low (non-native) to high (near-native), and the y-axis indicates the number of proteins in each bin.

the log-ratio color scale in Figure 2A and TM-score distribution in Figure 2B. The TM-score distribution reveals a roughly uniform count of target proteins from non-native to near-native regions. In contrast to prior benchmarks that focus on sampling near-native conformers around PDB structures, ProteinConformers achieves uniform coverage across the entire TM-score range.

### 3.5 Conformation Similarity Predictor

Unlike other datasets, ProteinConformers also provides energy metrics for each conformation, including TM–score, RMSD, and atomic-level energy scores from EvoEF2 [38], RW [40], RWplus [40], FoldX [37], and Rosetta [39]. To illustrate the utility of these features, we explore a challenging task: predicting structural similarity metrics without access to native structures using only energy features. The similarity metrics are TM–score and TM–RMSD. Here, TM-RMSD is defined as

$$\text{TM-RMSD} = \text{TM-score} + \frac{1}{1 + \text{RMSD}} \tag{1}$$

which ranges in $(0, 2]$, with higher values indicating greater structural similarity. Identical conformations (TM-score = 1, RMSD = 0) result in a TM–RMSD of 2. Predicting such global and local similarity metrics without reference structures is a nontrivial task. While AlphaFold's pLDDT score is commonly used for model quality estimation, its pTM metric is derived from Evoformer internals and is not directly accessible. We train a simple conformation similarity predictor using the ProteinConformers' energy features (Figure 3). The 87 proteins are split into train, test and validation dataset with ratio of 0.8, 0.1 and 0.1. We use AutoGluon[41] to fit a simple linear regression model using the five energy terms and protein length as features. Despite its simplicity, the model achieved promising results: PCC of 0.743 for TM–score and 0.748 for TM–RMSD, and SCC of 0.776 and 0.745, respectively. These findings demonstrate that ProteinConformers' energy features are valuable resource for conformational analysis and quality prediction.

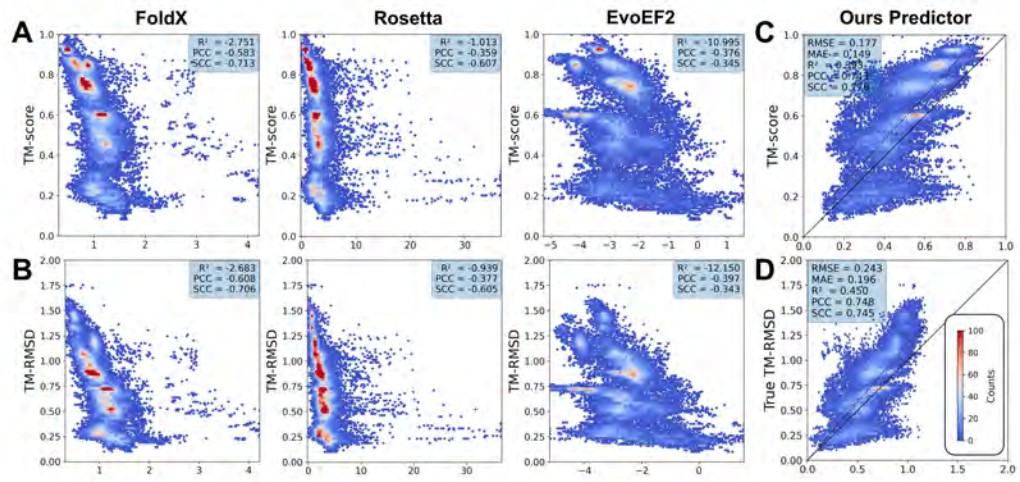

Figure 3: **Structural-quality assessment using classical energy functions and our learned Conformation Similarity Predictor.** (A) Density scatter plots of length-normalized potential energy (x-axis) versus TM-score (y-axis) for three classical scoring functions[37, 38, 39]. Color denotes point density; insets report $R^2$, Pearson correlation coefficient (PCC), and Spearman rank coefficient (SCC). (B) Analogous density plots of length-normalized potential energy (x-axis) versus TM-RMSD (y-axis). (C) True TM-score (y-axis) versus our predictor's estimated TM-score (x-axis). Inset statistics include RMSE, MAE, $R^2$, PCC, and SCC; the diagonal line denotes perfect agreement. (D) True TM-RMSD (y-axis) plotted against predicted TM-RMSD (x-axis).

## 4 Experiment

### 4.1 Evaluation Metrics and Compared Models

Based on the ProteinConformers dataset, we employ a coverage-based metric derived from free energy landscapes and introduce a new quantitative measure to assess two distinct aspects of the generated conformational ensembles: (i) **Coverage-based metrics**, which assesses the diversity of the protein conformational landscape by calculating the coverage rate of the generated conformers to the benchmark dataset, and (ii) **Protein Conformation Plausibility Map (PCPM)** and its derived **Protein Conformation Plausibility Score (PCPS)**, which evaluate the plausibility of the generated conformational ensembles in relation to known folding landscapes. To gain new insights into current conformational generative models, six different models were systematically compared. Two distilled variants of AlphaFlow were included: $AlphaFlow_{PDB}^{Dis}$, trained on experimental ensembles from PDB, and $AlphaFlow_{MD}^{Dis}$, trained on 300 K all-atom explicit-solvent MD trajectories. The same protocol is applied to $ESMFlow_{PDB}^{Dis}$ and $ESMFlow_{MD}^{Dis}$. For ESMDiff, the DDPM sampling paradigm with a step size of 1,000 is used. BioEmu is executed according to its official workflow. Each model generates 3,000 conformations for a curated subset of 18 proteins from the ProteinConformers.

#### 4.1.1 Protein Conformational Landscape Diversity

Protein conformational landscape diversity is measured by the overlap of low-energy regions in free energy landscapes between ProteinConformers and generated ensembles (Figure 4). We project protein conformations onto a 2D space using Principal Component Analysis (PCA) via MDAnalysis [42, 43], using the first two principal components from ProteinConformers for each protein. Generated conformers are projected onto this same space. The 2D projections are discretized into a $64 \times 64$ grid. Conformer density in each bin constructs free energy landscapes ($pmf = -kT \log(hist)$, where $hist$ is normalized density and $kT = 2.494$ kJ/mol, more in supplementary materials). Area intersection, coverage, and Jaccard index (equations in Supplementary Material) quantify low-energy region overlap between generation and ProteinConformers.

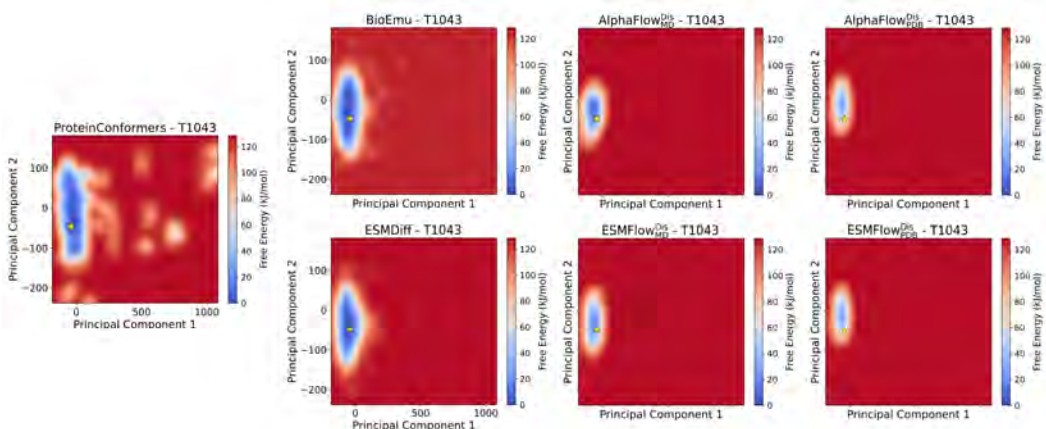

Figure 4: Case of free energy landscapes comparison from ProteinConformers and generative models.

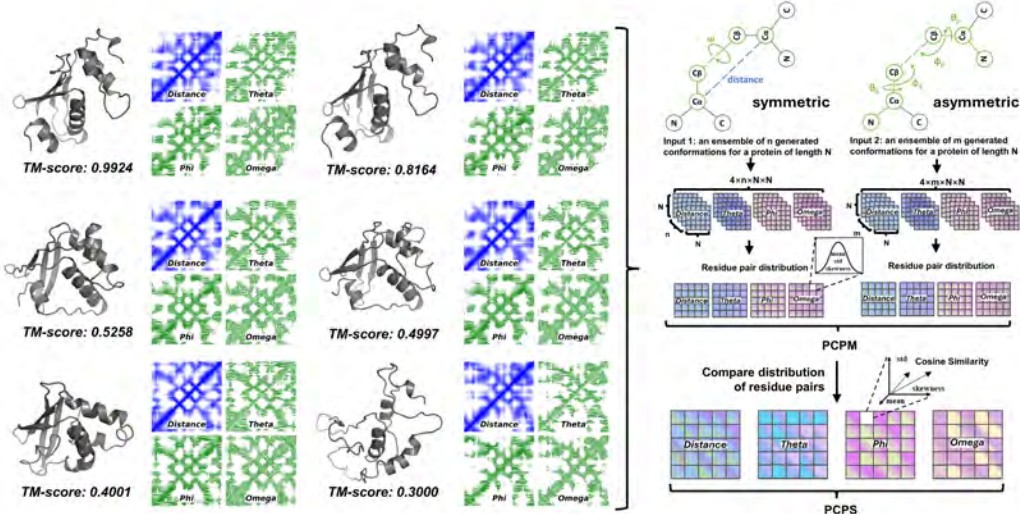

Figure 5: **Illustration of PCPM and PCPS.** Left: Six representative protein conformations with varying TM-scores and PCPMs. Upper Right: Geometric features construction: symmetric (distance $D$, dihedral $\Omega$) and asymmetric (dihedral $\Theta$, planar angle $\Phi$). Lower Right: PCPM and PCPS construction workflow, comparing residue-pair distribution statistics (mean, std. dev., skewness) of an input ensemble's PCPM with a benchmark PCPM via cosine similarity.

### 4.1.2 Protein Conformation Plausibility Maps and Scores

Traditional metrics such as TM-score [7], compare a generated structure with a single native reference and therefore quantify *structural similarity* rather than *atomic-level plausibility of the conformational distribution*. To address this gap, we introduce the PCPM and its derived PCPS, which directly assess the local physical realism of an ensemble of conformations (Figure 5 ) .

**PCPM Construction** For a protein of length $N$ and an ensemble of $M$ conformations $\mathcal{C} = \{C^{(m)}\}_{m=1}^M$, we compute four inter-residue geometric feature maps $G \in \{D, \Omega, \Theta, \Phi\}$ for every conformation, analogous to trRosetta [44] and DeepPotential [45]. Feature definitions are in Table 1.

For every residue-pair $(i, j) \in \{1, \ldots, N\}$ of each conformation $m$ in ensembles, we calculate the four geometric features. We then summarize the three central moments of the ensembles geometric features—mean ($\mu$), standard deviation ($\sigma$), and skewness ($\gamma$), which results representation tensor for geometric feature map $G$ is $\mathbf{P}^G \in \mathbb{R}^{3 \times N \times N}$, where $\mathbf{P}_{ij}^G \in \{\mu, \sigma, \gamma\}$. The complete PCPM is the concatenation $\mathbf{P} = \left[\mathbf{P}^D, \mathbf{P}^\Omega, \mathbf{P}^\Theta, \mathbf{P}^\Phi\right] \in \mathbb{R}^{12 \times N \times N}$.

Table 1: Geometric features used for PCPM. For $\Omega$, $\Theta$, and $\Phi$, a pseudo $C_\beta$ is used for glycine.

| Symbol | Type | Atoms Involved | Symmetry | Description |
|--------|------|----------------|----------|-------------|
| $D$ | Euclidean Dist. | $C_{\alpha,i}, C_{\alpha,j}$ | Symmetric | Distance ($D_{ij} = D_{ji}$) |
| $\Omega$ | Dihedral Angle | $C_{\alpha,i}, C_{\beta,i}, C_{\beta,j}, C_{\alpha,j}$ | Symmetric | Rotation about $C_{\beta,i} - C_{\beta,j}$ virtual axis |
| $\Theta$ | Dihedral Angle | $N_i, C_{\alpha,i}, C_{\beta,i}, C_{\beta,j}$ | Asymmetric | Orientation of $C_{\beta,j}$ to residue $i$ backbone |
| $\Phi$ | Planar Angle | $C_{\alpha,i}, C_{\beta,i}, C_{\beta,j}$ | Asymmetric | Position of $C_{\beta,j}$ to $C_{\alpha,i} - C_{\beta,i}$ bond |

**PCPS Computation** Let $\mathbf{P}^{PC}$ be ProteinConformer's PCPM and $\mathbf{P}^B$ be the benchmark method's map. After aligning residue indices (discarding missing ones), for every entry $(i, j)$ we build vectors $\mathbf{u}_{ij} = \left[\mu, \sigma, \gamma\right]^\top_{(PC)}$ and $\mathbf{v}_{ij} = \left[\mu, \sigma, \gamma, \right]^\top_{(B)}$, and compute cosine similarity $s_{ij} = \frac{\mathbf{u}_{ij}^\top \mathbf{v}_{ij}}{\|\mathbf{u}_{ij}\| \|\mathbf{v}_{ij}\|}$. Then PCPS $= \frac{1}{|\mathcal{C}|} \sum_{(i,j) \in \mathcal{C}} s_{ij}$, where $\mathcal{C}$ is the set of aligned residue-pair entries. PCPS $\in [-1, 1]$; higher values indicate better reproduction of atomic-level statistics of the authentic folding landscape.

We also used **Jensen-Shannon Divergence (JSD)** to evaluate the diversity, which measures the similarity between two probability distributions, $P$ and $Q$.

$$\text{JSD}(P\|Q) = \frac{1}{2} D_{KL}(P\|M) + \frac{1}{2} D_{KL}(Q\|M), \text{ where } M = \frac{1}{2}(P + Q)$$

Specifically, we employ two JSD-based metrics to assess the structural diversity and compactness distributions.

**JS Divergence of Pairwise Distances (JS-PwD):** This metric measures the similarity of inter-residue distance distributions. For each C$\alpha$–C$\alpha$ atom pair $(i, j)$ separated by at least 3 residues, we treat it as a distinct "channel". For each channel, we collect the distance values across all conformations in generated ensembles and ProteinConformers to form two distributions. We then compute the JSD for each channel and report the final JS-PwD as the mean JSD across all channels.

**JS Divergence of Radius of Gyration (JS-Rg):** This metric measures the similarity of overall molecular compactness between generated ensembles and ProteinConformers. For each ensemble, we compute the Radius of Gyration ($R_g$) of all conformations, build normalized histograms, and calculate the Jensen–Shannon divergence between the two distributions.

**Ramachandran outlier rate (Rama outlier %):** This metric quantifies the fraction of residues whose backbone dihedral angles ($\phi$, $\psi$) fall outside the favored Ramachandran regions. The final value is the mean outlier percentage across all conformations in the ensemble.

### 4.2 Evaluation Results

#### 4.2.1 Protein Conformational Landscape Diversity

Figure 4 illustrates the two-dimensional free energy landscapes for protein T1043 as derived from the ProteinConformers dataset and five generative models: AlphaFlow$_{MD}^{Dis}$, AlphaFlow$_{PDB}^{Dis}$, ESMFlow$_{MD}^{Dis}$, and ESMFlow$_{PDB}^{Dis}$, and BioEmu. Low-energy basins (in blue) indicate regions of high conformer density, while high-energy regions (in red) are sparsely populated. The yellow star marks the native structure projected onto the same PCA space. ProteinConformers exhibits a more diverse and widely distributed free energy landscape. In contrast, generative models, especially the distilled variants, tend to produce narrower distributions.

Table 2 compares three free energy overlap metrics—interaction, coverage, and Jaccard index—across energy thresholds of 5, 10, and 20 kJ/mol. BioEmu performs best at 5 kJ/mol with strong interaction and Jaccard scores, indicating effective low-energy sampling. In contrast, distilled models like AlphaFlow$_{MD}^{Dis}$ and ESMFlow$_{MD}^{Dis}$ show consistently lower scores, suggesting narrower sampling near dominant energy basins.

ESMDiff, trained solely on PDB data without MD simulations, still shows strong conformational exploration ability. This strength likely arises from its ESM3 foundation, pretrained on large-scale multimodal data, where massive sequence corpora provide implicit priors for sampling realistic conformations. Similar to PLAID [46], this highlights the potential of sequence-based pretraining for generating diverse, physically plausible protein structures.

Table 2: Protein Conformational Landscape with Different Energy Thresholds

| Method | Energy Interaction(↑) | | | Energy Coverage(↑) | | | Energy Jaccard(↑) | | |
|---|---|---|---|---|---|---|---|---|---|
| | 5 | 10 | 20 | 5 | 10 | 20 | 5 | 10 | 20 |
| AlphaFlow$_{MD}^{Dis}$ | 4.42 | 38.20 | 86.96 | 0.059 | 0.132 | 0.195 | 0.008 | 0.028 | 0.063 |
| AlphaFlow$_{PDB}^{Dis}$ | 4.65 | 41.26 | 89.66 | 0.064 | 0.152 | 0.196 | 0.012 | 0.046 | 0.086 |
| ESMFlow$_{MD}^{Dis}$ | 2.85 | 46.04 | 99.08 | 0.049 | 0.144 | 0.202 | 0.004 | 0.030 | 0.064 |
| ESMFlow$_{PDB}^{Dis}$ | 2.39 | 32.92 | 76.25 | 0.043 | 0.121 | 0.171 | 0.005 | 0.025 | 0.055 |
| AFsample2 | 1.14 | 13.86 | 90.43 | 0.004 | 0.047 | 0.116 | 0.003 | 0.020 | 0.070 |
| AlphaFold3 | 7.47 | 60.00 | 104.59 | 0.039 | 0.081 | 0.126 | 0.021 | 0.045 | 0.076 |
| BioEmu | 5.24 | 38.55 | 73.68 | 0.054 | 0.120 | 0.143 | 0.017 | 0.044 | 0.075 |

*Note:* The unit of the different energy thresholds is kJ/mol.

Our experimental results also show that the diffusion-based AlphaFold3 performs comparably to AlphaFlow, consistent with their similar generative paradigms. In contrast, AFsample2 exhibits the weakest performance, indicating that such perturbations only induce minor local structural variations and fail to capture global conformational diversity.

### 4.2.2 Protein Conformational Landscape Plausibility

The six generative models' PCPM is denoted as PCPM$^B$. We also computed three reference PCPM values on the ProteinConformers dataset itself: PCPM$^{near-native}$, using decoys with TM-score $\geq 0.5$; and PCPM$^{all}$, using the full set of decoys regardless of TM-score. We then compare PCPM$^B$ against each of the PCPM$^{near-native}$ and PCPM$^{all}$ to obtain the PCPS, which reflects the physical realism of the generative models in different regions of the protein conformational landscape—non-native, near-native, and global. The results are summarized in Table 3.

Table 3: Performance of PCPS with benchmark models

| | Near-native( ↑) | | | | | All (↑) | | | | |
|---|---|---|---|---|---|---|---|---|---|---|
| | D | O | P | T | A | D | O | P | T | A |
| AlphaFlow$_{MD}^{Dis}$ | 0.946 | 0.627 | 0.801 | 0.537 | 0.728 | 0.900 | 0.644 | 0.772 | 0.574 | 0.723 |
| AlphaFlow$_{PDB}^{Dis}$ | 0.922 | 0.616 | 0.786 | 0.564 | 0.722 | 0.871 | 0.592 | 0.746 | 0.542 | 0.688 |
| ESMFlow$_{MD}^{Dis}$ | 0.957 | 0.604 | 0.811 | 0.539 | 0.728 | 0.922 | 0.710 | 0.814 | 0.665 | 0.778 |
| ESMFlow$_{PDB}^{Dis}$ | 0.859 | 0.516 | 0.710 | 0.468 | 0.638 | 0.819 | 0.603 | 0.700 | 0.578 | 0.675 |
| BioEmu | 0.967 | **0.717** | **0.826** | **0.665** | **0.794** | 0.935 | 0.764 | **0.862** | **0.713** | **0.818** |
| ESMDiff | **0.971** | 0.649 | 0.693 | 0.191 | 0.626 | **0.967** | **0.807** | 0.639 | 0.338 | 0.688 |

*Note:* Metrics abbreviations: D-Distance, O-Omega, P-Phi, T-Theta, A-Average.

Table 4: Comparison of models on JS metrics and Ramachandran outliers

| Model | Avg. JS-PwD (↓) | Avg. JS-Rg (↓) | Rama Outlier % (↓) |
|---|---|---|---|
| AlphaFlow$_{MD}^{Dis}$ | 0.424 | 0.310 | 13.43% |
| AlphaFlow$_{PDB}^{Dis}$ | 0.445 | 0.412 | 9.93% |
| ESMFlow$_{MD}^{Dis}$ | 0.283 | 0.507 | 13.36% |
| ESMFlow$_{PDB}^{Dis}$ | 0.417 | 0.697 | 12.26% |
| AFsample2 | 0.490 | 0.311 | 6.61% |
| AlphaFold3 | 0.632 | 0.293 | **2.66%** |
| BioEmu | 0.235 | 0.206 | 10.69% |
| ESMDiff | **0.128** | **0.104** | 20.41% |

From Table 3, MD-trained AlphaFlow and ESMFlow outperform their PDB-trained counterparts across most metrics. While all models align well with native Cα distance distributions, they reproduce orientation angles poorly. This may stem from FAPE loss focusing on per-residue alignment without enforcing inter-residue pose consistency, and from two-stage architectures that limit torsional realism.

BioEmu offers the best overall physical plausibility, whereas ESMDiff, despite its superior distance modeling, still lacks angular accuracy.

We evaluated geometric and stereochemical quality across models Table4. ESMDiff achieved the best geometric similarity (lowest JS-PwD = 0.128, JS-Rg = 0.104) but had the highest Ramachandran outlier rate (20.41%). AlphaFold3 showed the most stereochemically plausible structures with only 2.66% outliers, while AFsample2 and BioEmu performed moderately. Diffusion- and flow-based variants exhibited intermediate to high outlier levels.

These results reveal a trade-off between geometric similarity and physical plausibility. Models optimized for distributional alignment with experimental ensembles tend to sacrifice stereochemical accuracy, while structure-prediction-based frameworks such as AlphaFold3 and AFsample2 maintain physically realistic torsion angles but explore a narrower conformational space. The consistency between Ramachandran statistics and PCPS trends reinforces the reliability of our plausibility assessment framework.

# 5 Discussion

This study introduces ProteinConformers, a large-scale, diverse and physically realistic benchmark dataset with corresponding evaluation metrics. We collect over 40,387 decoys from 87 CASP targets, refine each decoy with all-atom MD simulations and sample 381,546 conformations at a cost exceeding 6 million CPU hours. We further defined a dual-axis evaluation framework comprising conformational diversity and physical plausibility.

We applied ProteinConformers and corresponding evaluation metrics to benchmark generative models. Our results show that models trained on native structures excel at exploring near-native energy funnel, whereas those trained on MD trajectories yield superior atomic-level physical realism. Although all methods reproduce good inter-residue distance distributions, they underperform in sampling inter-residue torsion angles. We believe that integrating both PDB and MD data and focusing on torsion-angle learning will be crucial for improving conformational plausibility.

We observe that current multi-conformer generators perform poorly on torsion-angle prediction, likely due to weak coupling between local geometry and long-range constraints. To address this, we suggest three complementary directions: introducing long-range orientation modules to propagate angular dependencies across residues; incorporating joint supervision of backbone and side-chain torsions to enhance stereochemical realism; and applying energy-based angular regularization to provide physics-guided priors that improve plausibility without compromising diversity.

ProteinConformers has several limitations. Short MD simulations may miss high-energy conformations, and the decoy set lacks fragment assembly or physics-based diversity. The similarity predictor is preliminary and unvalidated at scale. Due to computational limits, only 3,000 conformations per protein were generated, preventing full model benchmarking. Dataset non-overlap hinders direct comparison, and CASP-derived seeds may bias landscape coverage. Short MD windows miss rare transitions, limiting Boltzmann sampling. The absence of enhanced-sampling methods reduces exploration of high-energy states, and generalizability to experimental ensembles remains untested. Plausibility metrics, such as Ramachandran outliers, may also be sensitive to parameter choices.

Future work will diversify decoy generation pipelines by combining fragment assembly, coarse-grained physics simulations and generative modeling for more systematic coverage. We will also extend the MD sampling protocol by integrating enhanced sampling techniques and longer trajectory. Finally, we will develop an online platform that provides data browsing, metric computation and model evaluation in a unified environment, facilitating use by the computational biology and drug design communities.

# 6 Acknowledgements

This work is supported in part by Singapore Ministry of Education (T1 251RES2309) and National University of Singapore Startup Grants (A-8001129-00-00, A-0009651-30-00).

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

# A   Technical Appendices and Supplementary Material

## A.1   Equations for Conformational Energy Landscape Overlap Analysis

To quantify the similarity between the protein conformations generated by AI-based models and those in the ProteinConformers dataset, the following three commonly used overlap metrics are employed: Interaction overlap, coverage, and the Jaccard index. These metrics evaluate the extent of agreement in low-energy regions between the protein conformers from different models of the same protein, based on a specified energy threshold.

Let $A = \{A_{i,j}\}$ and $B = \{B_{i,j}\}$ where $i, j \in [0, N]$, denote the two-dimensional free energy landscapes corresponding of two conformational ensembles. Each element $A_{i,j}$ and $B_{i,j}$ represents the free energy value at a specific grid point in the conformational energy landscape. For a given energy threshold $\tau$ (e.g., 40 kJ/mol), the number of shared low-energy conformations is defined as:

$$|A \cap B| = \sum_{i,j=1}^{N} \mathbf{1}[A_{i,j} < \tau \wedge B_{i,j} < \tau]$$

where $N = 63$, and $\mathbf{1}[\cdot]$ is the indicator function, which returns 1 if the condition inside is true and 0 otherwise.

The low energy area of different conformational free energy landscape under different threshold are given by:

$$|A| = \sum_{i,j=1}^{N} \mathbf{1}[A_{i,j} < \tau], \qquad |B| = \sum_{i,j=1}^{N} \mathbf{1}[B_{i,j} < \tau]$$

Using the above definitions, the overlap metrics are computed as follows:

- **Interaction**:

$$\text{Interaction} = |A \cap B|$$

- **Coverage** (proportion of low-energy conformations in $A$ also found in $B$):

$$\text{Coverage} = \frac{|A \cap B|}{|A|}$$

- **Jaccard Index** (symmetric overlap metric between both sets):

$$\text{Jaccard} = \frac{|A \cap B|}{|A| + |B| - |A \cap B|}$$

## A.2   Free Energy Landscapes Comparison

This section provides additional figures comparing the conformational landscapes of protein conformers from ProteinConformers with those generated by AI models.

## A.3   Overview the ProteinConformers proteins

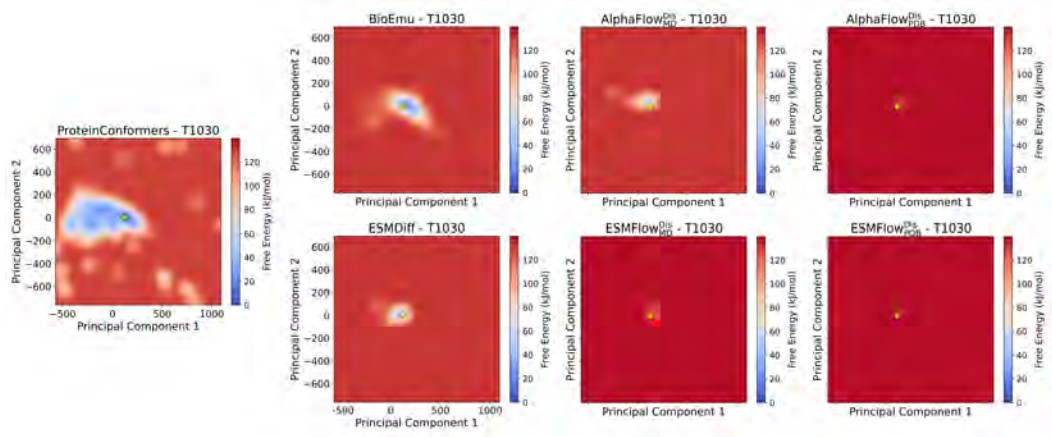

(a) Comparison of 2D conformational landscapes.

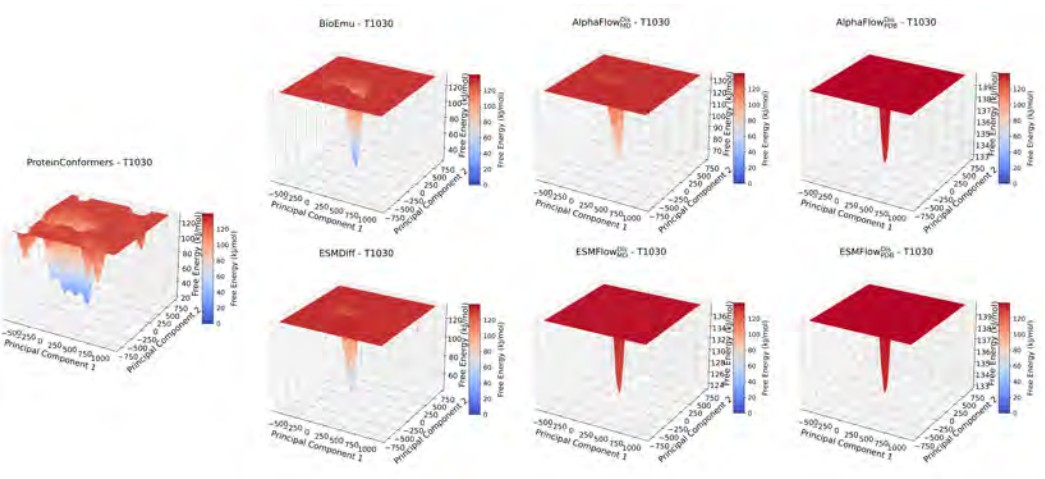

(b) Comparison of 3D conformational landscapes.

Figure 6: Comparison of conformational landscapes for protein T1030, generated by ProteinConformers and protein conformation generative models.

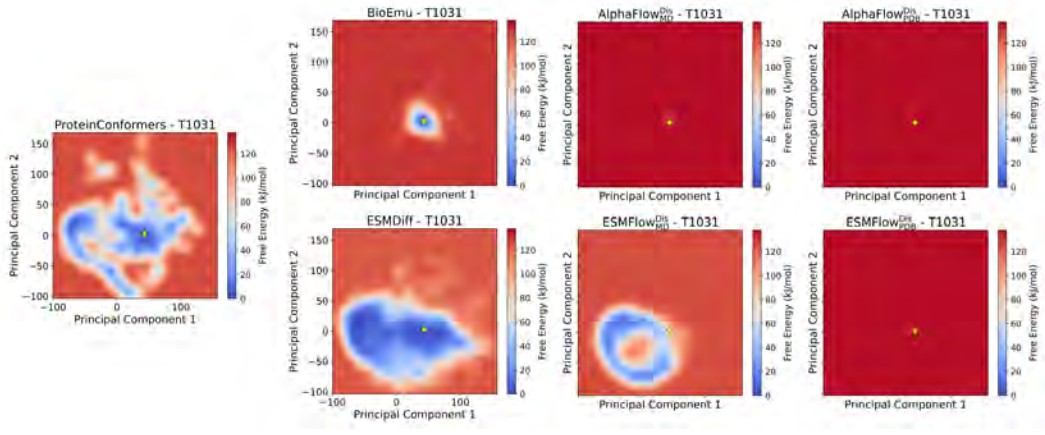

(a) Comparison of 2D conformational landscapes.

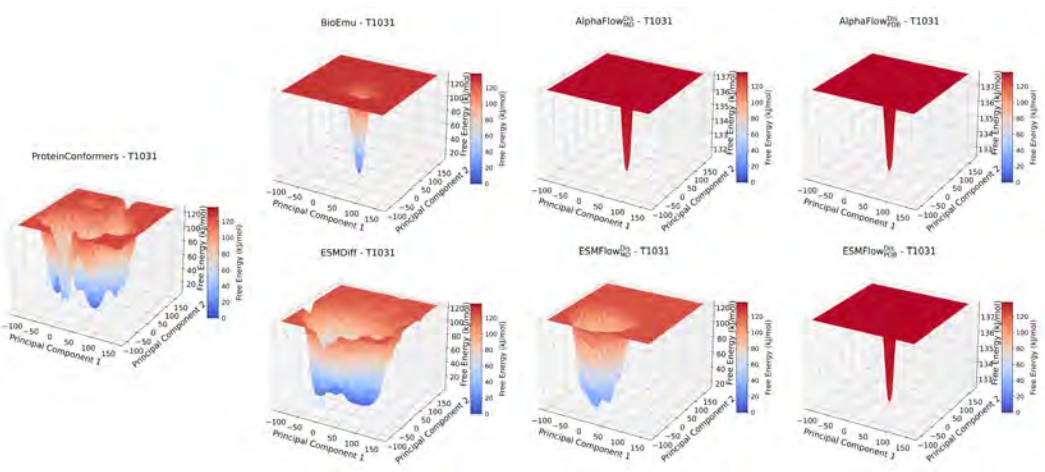

(b) Comparison of 3D conformational landscapes.

Figure 7: Comparison of conformational landscapes for protein T1031, generated by ProteinConformers and protein conformation generative models.

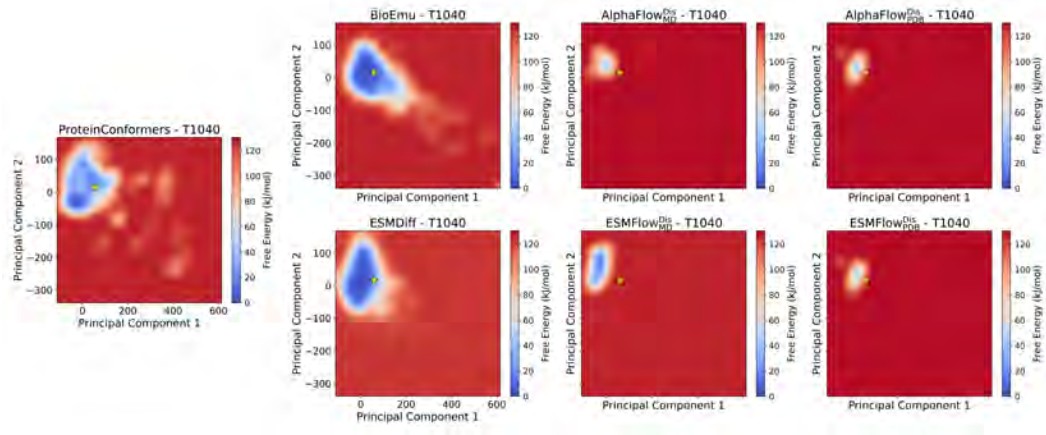

(a) Comparison of 2D conformational landscapes.

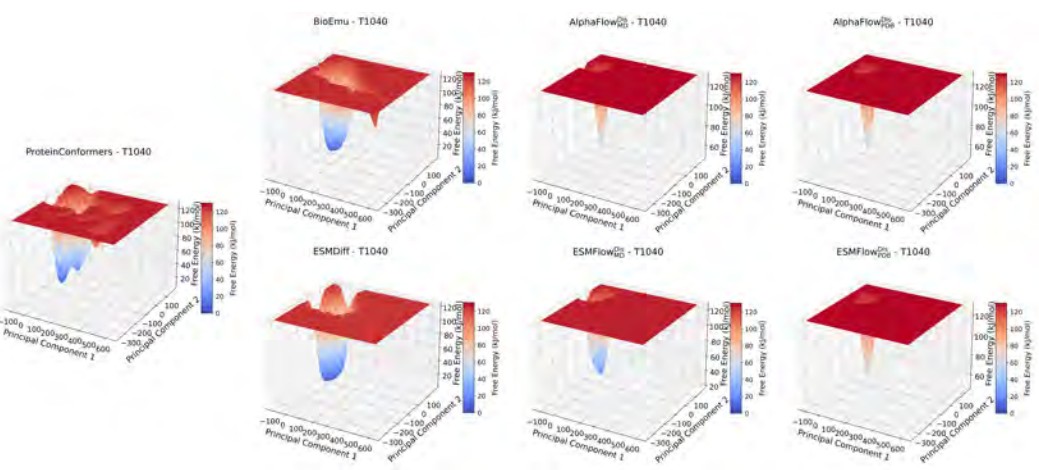

(b) Comparison of 3D conformational landscapes.

Figure 8: Comparison of conformational landscapes for protein T1040, generated by ProteinConformers and protein conformation generative models.

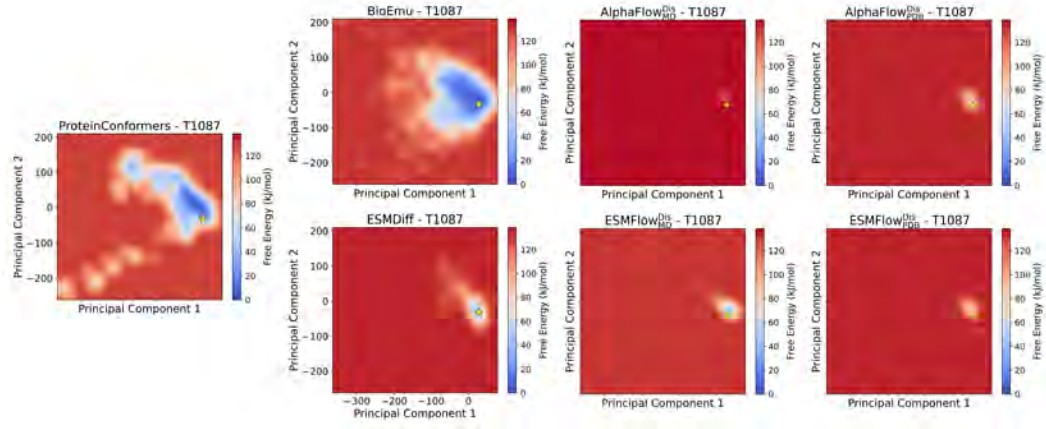

(a) Comparison of 2D conformational landscapes.

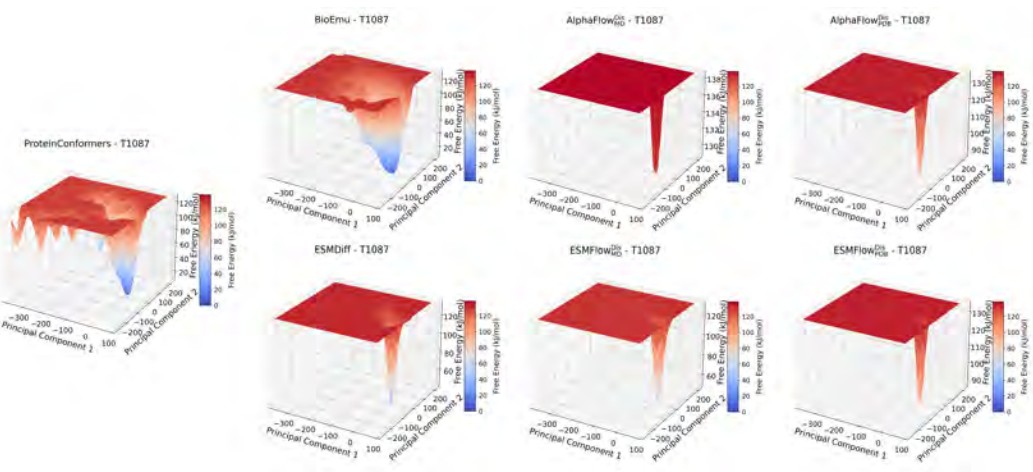

(b) Comparison of 3D conformational landscapes.

Figure 9: Comparison of conformational landscapes for protein T1087, generated by ProteinConformers and protein conformation generative models.

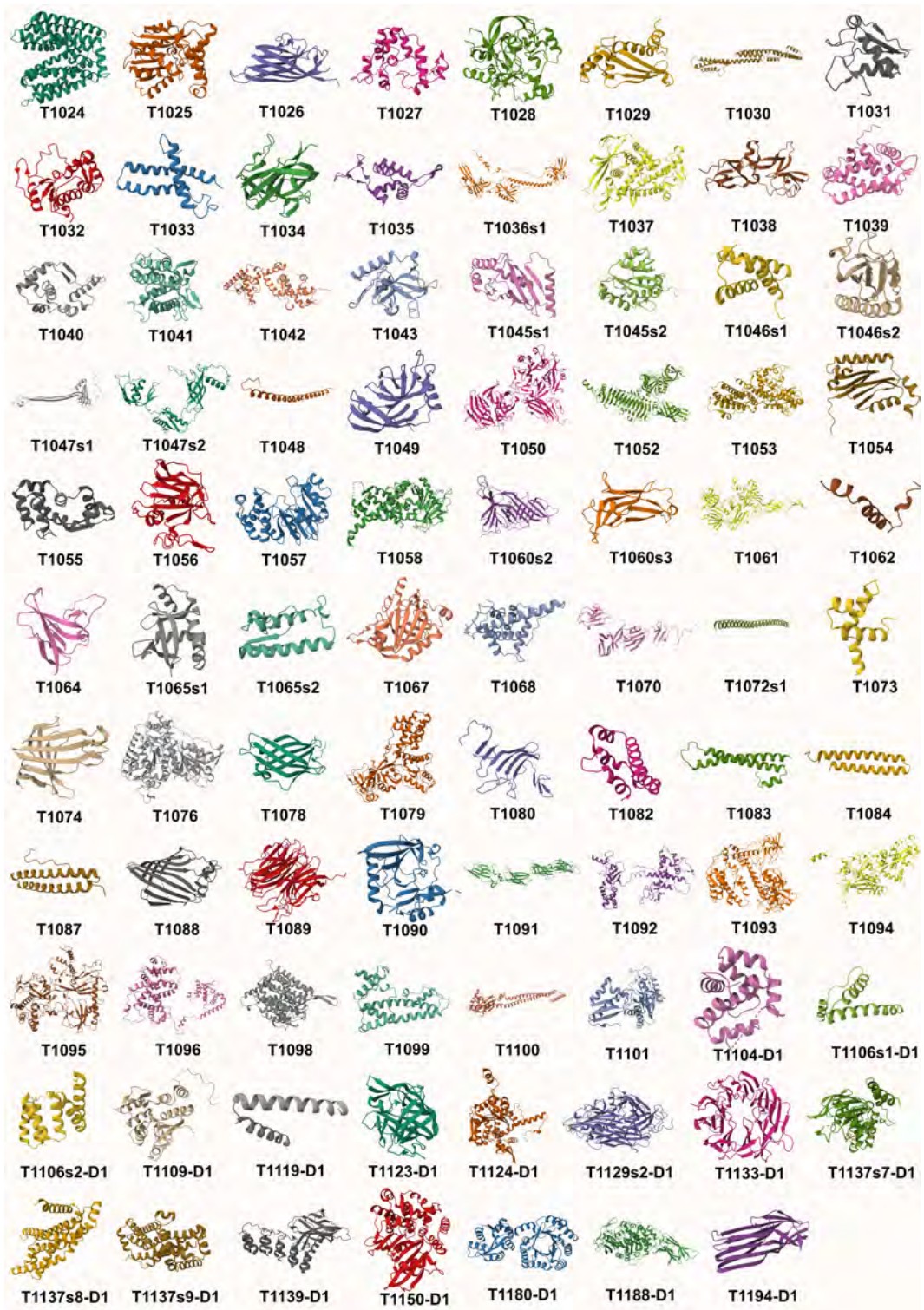

Figure 10: The 3D native structures of all 87 proteins in ProteinConformers.