# OpenReview forum: "ProteinConformers: Benchmark Dataset for Simulating Protein Conformational Landscape Diversity and Plausibility"
_NeurIPS.cc/2025/Datasets_and_Benchmarks_Track — NeurIPS 2025 Datasets and Benchmarks Track poster_

### Official Review · Reviewer_gtWq · 2025-06-19

**Rating:** 5
**Confidence:** 5

**Summary:**

This paper introduces ProteinConformers, a large-scale benchmark dataset designed to evaluate generative models of protein conformational landscapes. The authors aggregate 40,000 decoy structures across 87 CASP targets and perform molecular dynamics (MD) refinement to generate over 380,000 high-quality atomistic conformations. A two-axis evaluation framework is proposed: one axis quantifies conformational diversity (via free energy landscape coverage), and the other quantifies plausibility (via distance and angle-based structural criteria). Several recent structure-generating models are benchmarked, revealing that PDB-trained models (e.g., ESMDiff) show broader exploration capacity than MD-trained counterparts.

**Dataset Code Accessibility:**

Yes

**Ethical Considerations:**

No, there are no or only very minor ethics concerns

**Limitations Weaknesses:**

Angle-based plausibility metrics could be improved: RMSD of torsions may not fully capture stereochemical realism. Consider including metrics such as Ramachandran outlier rates.

While simulations are efficient and large-scale, the relatively short MD durations (125–375 ps) might miss high-energy conformers or rare transitions, potentially underrepresenting the true conformational landscape.

Despite excellent inter-residue distance predictions by most models, torsional angle sampling remains poor. However, the paper does not propose or explore model design adjustments to specifically address this deficiency.

**Strengths Contributions:**

Well-motivated and timely: Addresses the growing need to move beyond static structure prediction.
Data scale and quality: The dataset is physically sound, diverse, and curated with computationally intensive MD refinement.

Six recent generative models are benchmarked, with insights into their performance trade-offs.

---

> ### Author Rebuttal · Authors · 2025-07-31
>
> **Dear Reviewer gtWq,**
>
> We sincerely thank the reviewer for the encouraging and thoughtful feedback. We greatly appreciate your recognition of the motivation, scale, and quality of the dataset, as well as the relevance of our proposed evaluation framework and the benchmark analysis. Your positive assessment of the timeliness of our work, the rigor of the MD-based data curation, and the comparative evaluation of recent generative models is deeply appreciated. We also value your constructive suggestions, in response, we have clarified these points in the revised manuscript and discussed possible directions for future work. Below, we provide detailed responses to your comments item by item.
>
> ---
>
> **[W1] (Limitations Weaknesses)** *Angle-based plausibility metrics could be improved: RMSD of torsions may not fully capture stereochemical realism. Consider including metrics such as Ramachandran outlier rates.*
>
> **[A1]** We fully agree with the reviewer’s suggestion. Ramachandran outlier rates are indeed a meaningful metric for evaluating backbone torsion angle plausibility. We computed outlier rates by applying PyRosetta’s Ramachandran scoring function to all residues (excluding termini) in each conformation. Residues with φ/ψ angles outside the allowed regions were marked as outliers. Importantly, we extended the analysis to eight methods (as opposed to six in the original paper), including AlphaFold3 and AFSample2 to provide a more comprehensive benchmark of the AlphaFold series. The results are consistent with the trends reported by PCPS, supporting its reliability as an angle-based metric. AlphaFold3 showed the lowest outlier rate (lower is better), followed by AFSample2. Other multi-conformer samplers based on fine-tuned structure prediction models had higher outlier rates. Among them, BioEmu performed better than AlphaFlow and ESMFlow, while ESMDiff had the highest outlier rate, which aligned with PCPS trends. More details will be provided in revision article. We appreciate the reviewer’s helpful suggestion and will include detailed results in the revised paper. This addition further supports the conclusions of our work.
>
>
> **[W2] (Limitations Weaknesses)** *While simulations are efficient and large-scale, the relatively short MD durations (125–375 ps) might miss high-energy conformers or rare transitions, potentially underrepresenting the true conformational landscape.*
>
> **[A2]** We sincerely thank the reviewer for this important observation. We fully agree that short MD trajectories may fail to capture rare transitions and high-energy intermediate states. Achieving this level of sampling would likely require simulations on the millisecond scale, which is currently infeasible given the size of our dataset. Therefore, we initialized simulations from a broad set of structurally diverse decoys, including CASP submissions and physics-based 3DRobot-sampled conformers. This strategy increases the chances of reaching varied regions of the conformational space, even within short MD windows. Nevertheless, we acknowledge that this approach may still not able to sample all conformations. As we stated in section Dissussion “The relative short MD simulations may miss high-energy conformations”. In future work, we will use more systematic sampling methods (e.g., metadynamics) to broaden the sampled landscape. We appreciate the reviewer’s suggestion and will explicitly emphasize in the revised discussion that future improvements could involve protein-agnostic enhanced sampling protocols, such as metadynamics or replica exchange, to more thoroughly explore rare and high-energy conformational states.
>
> **[W3] (Limitations Weaknesses)** *Despite excellent inter-residue distance predictions by most models, torsional angle sampling remains poor. However, the paper does not propose or explore model design adjustments to specifically address this deficiency.*
>
> **[A2]** We appreciate the reviewer for this insightful comment. As noted in section 4.2.2, we discussed possible causes, e.g., “FAPE loss… imposes no constraints on the relative poses of adjacent residues” and “two-stage pipelines… limit torsional angle distributions.” However, we agree it was an oversight not to offer actionable modeling strategies. In the final version, we will expand the section Discussion to suggest: (1) For Ab initio models training, adding long-range-aware modules to better capture residue-pair orientations; (2) For fine-tuning pretrained models, incorporating side-chain losses to improve fidelity; (3) For sampling enhancing models, introducing energy-based angular regularization for physics guidance. We hope these updates address the reviewer’s concern and enhance the practical utility of our benchmark.
>
> ---
>
> **We hope that our responses satisfactorily address your concerns. Should you have any further questions during the rebuttal period, we will be readily available to provide clarification.**

---

### Official Review · Reviewer_owsH · 2025-07-01

**Rating:** 5
**Confidence:** 4

**Summary:**

The paper introduces a new benchmark dataset for evaluating the protein conformer generation task. Conformer generation is a new frontier in protein generative models where the goal is to generate multiple conformations of the protein according to Boltzman distribution. However, it is challenging to evaluate the task since most existing datasets (like PDB) contain only a single conformer. This paper creates a new evaluation dataset by running large scale MD to obtain multiple conformers for CASP proteins. It then creates a set of novel evaluation metrics focusing on the coverage and likelihood. It evaluates several popular protein generative models and provides useful insights for the limitations of existing models.

**Dataset Code Accessibility:**

Partly

**Dataset Code Comments:**

The authors have released the benchmark dataset on huggingface. However, the evaluation code is not yet released.

**Ethical Considerations:**

No, there are no or only very minor ethics concerns

**Final Justification:**

I increased my score from 4 to 5 since the authors satisfactorily addressed most of my concerns. See my comments in more detail.

**Limitations Weaknesses:**

- The protein conformers are generated by aggregating all CASP submissions. Although MD simulations are performed, there is no guarantee that these conformers capture all the modes in the protein conformer landscape. There is also no guarantee that they follow the Boltzman distribution. This is the biggest limitation of the benchmark.
- The number of evaluated models is limited. For example, the authors didn’t evaluate AlphaFold2 / AlphaFold 3.

**Strengths Contributions:**

- The dataset contains multiple protein conformers for 87 CASP proteins. CASP proteins are large proteins so it is non-trivial to obtain multiple conformers. This dataset provides a timely benchmark to evaluate their protein conformer generation ability.
- Short MD simulations are performed to relax the structures of each protein to generate the dataset.
- The conformer landscape diversity metrics help to evaluate the coverage of generated conformers at different energy levels. The fine-grained insights is extremely useful to understand the conformer generation behavior for different models.
- The authors differentiate the near-native conformers and other conformers in the evaluation. Since the non-native conformers are likely lower in quality, it helps to improve the quality of the benchmark despite the difficulty of capturing all modes in protein conformer landscape.

---

> ### Author Rebuttal · Authors · 2025-07-31
>
> **Dear Reviewer, owsH,**
>
> We sincerely thank the reviewer for the thoughtful and detailed evaluation of our work. We greatly appreciate your recognition of the novelty and relevance of our benchmark dataset, the usefulness of our conformer diversity metrics, and the careful handling of near-native versus non-native structures. These points reflect core goals of our work, and we are encouraged that they were well-received. We also appreciate the constructive feedback regarding model coverage and assumptions about sampling fidelity. In response, we have revised the manuscript to clarify these limitations and have conducted new experiments to address them. Detailed responses to each of your comments are provided below.
>
> ---
>
> **[W1] (Limitations Weaknesses)** *The protein conformers are generated by aggregating all CASP submissions. Although MD simulations are performed, there is no guarantee that these conformers capture all the modes in the protein conformer landscape. There is also no guarantee that they follow the Boltzman distribution. This is the biggest limitation of the benchmark.*
>
> **[A1]** We sincerely thank the reviewer for pointing out this important limitation. In this work, MD is used to optimize to produce physically reasonable conformations with processes that include temperature and pressure equilibrium and short-range product MD simulation. We completely agree that short-term MD simulations cannot capture all modes in the protein conformational landscape, so we started with the CASP prediction structures and the 3DRobot sample structures to get the most possible access to the conformational landscape. Besides, we do not rely on structural clustering in conformational selection, which reduces our need for the Boltzmann distribution, although we believe that it should be subject to the Boltzmann distribution after a complete system preparation process. We also note that, given the astronomical size of protein conformational spaces, it is theoretically intractable to exhaustively sample all possible states at scale. This limitation is shared across many studies in this field. Following your suggestion, we believe that using more systematic method (e.g., metadynamics ) in the future can explore broader landscape. In the article, we stated “The decoy set does not incorporate much fragment assembly or physics-based methods to improve diversity” in the section Discussion, but we will further revise the article to improve the limitation discussion and highlight your suggestion as a key direction for future dataset refinement. We are grateful to the reviewer for identifying this crucial point.
>
> **[W2]  (Limitations Weaknesses)** *The number of evaluated models is limited. For example, the authors didn’t evaluate AlphaFold2 / AlphaFold 3.*
>
> **[A2]** Thank you for the constructive comment. To assess the performance of AlphaFold2 and AlphaFold3, we conducted a series of experiments. In addition, we evaluated AFSample2, a modified version of AlphaFold2 specifically designed for conformation ensemble generation, whereas the original AlphaFold2 primarily targets static structure prediction. We also included experiments to assess the conformational diversity generated by AlphaFold3, as its diffusion-based architecture introduces stochasticity that may enhance its ability to model multiple conformational states.
>
> In our experimental setup, we aimed to generate 3,000 conformations for each protein. However, the conformation generation process is computationally expensive. At present, we have successfully completed conformation generation for 14 proteins using AFsample2. Consequently, our current comparison of protein conformational landscapes is limited to these 14 proteins. We are still running the program to generate conformations of other proteins, and we will include the comparison with other proteins in the revised manuscript.
>
> |**Method** |Energy|**Interaction**|($\\uparrow$)|   |Energy|**Coverage**| ($\\uparrow$)|   |Energy|**Jaccard**|($\\uparrow$)|
> |------------|:---:|:---:|:---:|:---:|:---:|:---:|:---:|:---:|:---:|:---:|:---:|
> | | **5** | **10** | **20** | |**5** | **10** | **20** || **5** | **10** | **20** |
> | AlphaFlow$_{\\mathrm{MD}}^{\\mathrm{Dis}}$ | 2.83 | 59.39 | 108.06 || 0.029 | 0.130 | 0.178 || 0.005 | 0.039 | 0.072 |
> | AlphaFlow$_{\\mathrm{PDB}}^{\\mathrm{Dis}}$ | 6.44 | 64.17 | 115.28 || 0.043 | 0.136 | 0.179 || 0.012 | 0.047 | 0.086 |
> | ESMFlow$_{\\mathrm{MD}}^{\\mathrm{Dis}}$ | 2.33 | 76.33 | 143.72 || 0.022 | 0.143 | 0.237 || 0.004 | 0.043 | 0.089 |
> | ESMFlow$_{\\mathrm{PDB}}^{\\mathrm{Dis}}$ | 6.44 | 64.17 | 115.28 || 0.043 | 0.136 | 0.179 || 0.012 | 0.046 | 0.086 |
> | AlphaFold3 | 7.47 | 60.00 | 104.59 || 0.039 | 0.081 | 0.126 || 0.021 | 0.045 | 0.076 |
> | AFsample2 | 1.14 | 13.86 | 90.43 || 0.004 | 0.047 | 0.116 || 0.003 | 0.020 | 0.070 |
> | BioEmu | **13.00** | 91.11 | 146.78 || 0.079 | 0.178 | 0.250 || **0.031** | 0.089 | 0.138 |
> | ESMDiff | 10.12 | **147.88** | **225.29** || **0.148** | **0.316** | **0.336** || 0.023 | **0.107** | **0.166** |
>
> Current results indicate that the diffusion-based AlphaFold3 achieves performance comparable to AlphaFlow, which utilize flow-matching techniques for conformation generation. AlphaFlow, fine-tuned from AlphaFold2, introduces randomness through noise initialization to produce diverse conformations. Since AlphaFold3 and AlphaFlow share highly similar training paradigms and rely on related generative principles, their overall performance metrics naturally converge to a similar level.
>
> The experimental results further demonstrate that AFsample2, which generates conformations by randomly masking columns in the MSA, exhibits the weakest performance among the evaluated methods. This finding suggests that relying solely on column masking within the MSA provides only a limited ability to explore the broader conformational landscape of a protein. Since the MSA encodes evolutionary relationships across sequence positions, masking its columns primarily perturbs local sequence information. As a result, this approach tends to sample conformations restricted to local structural variations around the reference fold, thereby failing to effectively capture the global diversity of the protein’s conformational space.
>
> **[W3]  (Dataset Code Comments)** *The authors have released the benchmark dataset on huggingface. However, the evaluation code is not yet released.*
>
> **[A3]** Thank you for your comment. We have included the evaluation code within the dataset repository under the file name ProteinConformers_code.zip. However, we acknowledge that its placement alongside the dataset files may cause inconvenience for users. To improve accessibility and usability, we will upload the complete codebase to a dedicated GitHub repository and provide the corresponding repository link in the final revised manuscript.
>
> ---
>
> **We sincerely thank you for your thoughtful and constructive feedback, which has been invaluable in improving our manuscript. We hope that our responses have satisfactorily addressed your comments and concerns. Should you have any further questions during the rebuttal period, we remain fully available to provide clarification and additional details as needed.**

---

> > ### Author Response · Authors · 2025-08-05
> >
> > Dear Reviewer owsH,
> >
> > We hope this message finds you well.
> >
> > We are writing to kindly follow up regarding the author rebuttal we submitted last week for Paper #1323. We truly appreciate the time and thought you invested in your initial review, and we have carefully addressed your comments and suggestions in our response.
> >
> > We understand that this is a particularly busy period for you, but if you have an opportunity to review our clarifications as you prepare your final recommendation, we would be sincerely grateful.
> >
> > Thank you very much for your time and effort in evaluating our work.
> >
> > Sincerely,
> > The Authors of Paper #1323

---

> > ### Comment · Reviewer_owsH · 2025-08-07
> >
> > I appeciate the authors for the detailed response and additional experiments. They've satisfactorily addressed most of my concerns. I therefore increase my score to 5. Before the camera ready version, I suggest the authors to upload their code to github and imporve the limitation discussion session.

---

> > > ### Author Response · Authors · 2025-08-08
> > >
> > > Dear Reviewer owsH:
> > >
> > > Thank you so much for your positive feedback and for your time in reviewing our detailed response and additional experiments.
> > >
> > > We are very grateful for your support and for increasing your score.
> > >
> > > We will certainly follow your excellent suggestions for the camera-ready version.
> > >
> > > We will upload our code to GitHub and will expand the limitation discussion as you recommended.
> > >
> > > Thank you again for your constructive guidance throughout this process.
> > >
> > > Best regards,
> > >
> > > The Authors of Paper #1323

---

### Official Review · Reviewer_NMAr · 2025-07-02

**Ethics Flags:** Data privacy, copyright, and consent,…
**Rating:** 4
**Confidence:** 5

**Summary:**

Existing protein conformation benchmark fail to capture the full energy landscape, limiting their ability to evauate the diversity and physical plausibility of AI-generated structures, the paper introduce ProteinConformers. The paper propose novel metrics to evaluate conformational diversity and plausibility, and systematically benchmark six protein conformation generative models. Extensive experiments show that the ProteinCoformers can enhance a model's ability to explore conformational space, potentially reducing influence on MD-derived data.

**Dataset Code Accessibility:**

Yes

**Ethical Considerations:**

No, there are no or only very minor ethics concerns

**Limitations Weaknesses:**

1. The ProteinCoformers  introuduces two evaluation metrics, which need to include more evaluation metircs.
2. The ProteinCoformers lacks more state-of-the-art methods to compare with the new methods.

**Strengths Contributions:**

1. The paper's presentation is well-written, organized, and easy to understand.
2. The paper collected and filtered over 40,000 decoy conformers generated by hunderds of different traditional and AI based prediction algorithms.

---

> ### Author Rebuttal · Authors · 2025-07-31
>
> Response to Reviewer NMAr
>
> We sincerely thank the reviewer for their thorough and thoughtful review of our work. In response, we have conducted substantial new experiments and analyses that we believe significantly strengthen the manuscript.
>
> **[W1]** *The ProteinCoformers introuduces two evaluation metrics, which need to include more evaluation metircs.*
>
> **[A1]** We fully agree that incorporating additional, well-established evaluation metrics leads to a more comprehensive assessment. In response, we have expanded our evaluation framework by introducing two key metrics from recent literature to quantify the distributional similarity between generated ($E_{gen}$) and ground-truth ($E_{gt}$) ensembles. These metrics are based on the Jensen-Shannon Divergence (JSD), which measures the similarity between two probability distributions, $P$ and $Q$. The comparisons of these metrics on different SOTA models can be found in the answer to the second question. It is a symmetrized version of the Kullback-Leibler (KL) divergence, defined as:
>
> $JSD(P||Q) = \frac{1}{2} D_{KL}(P||M) + \frac{1}{2} D_{KL}(Q||M)$, where $M = \frac{1}{2}(P+Q)$.
>
> 1.  **Jensen-Shannon (JS) Divergence of Pairwise Distances (JS-PwD):** This metric measures the similarity of inter-residue distance distributions.
>     * **Implementation:** For each Cα-Cα atom pair $(i, j)$ separated by at least 3 residues, we treat it as a distinct "channel". For each channel, we collect the distance values across all conformations in $E_{gen}$ and $E_{gt}$ to form two distributions. We then compute the JSD for each channel and report the final JS-PwD as the mean JSD across all channels.
>
> 2.  **JS Divergence of Radius of Gyration (JS-Rg):** This metric compares the distribution of overall molecular compactness.
>     * **Implementation:** We first calculate the Radius of Gyration ($R_g$) for every conformation in both $E_{gen}$ and $E_{gt}$. The resulting sets of scalar $R_g$ values are then binned into two normalized histograms, representing the probability distributions of molecular compactness for each ensemble. The final JS-Rg is the JSD between these two distributions.
>
> Our analysis using these new metrics reveals that models like **ESMDiff** and **BioEmu** consistently achieve the lowest (best) JS-PwD and JS-Rg scores, indicating their generated ensembles closely match the geometric distributions of our MD-derived ground truth. This new analysis will be detailed in the revised manuscript.
>
>
>
> **[W2]** *The ProteinCoformers lacks more state-of-the-art methods to compare with the new methods.*
>
> **[A2]** We fully agree that benchmarking against the latest methods is essential. In response to this valuable suggestion, we have devoted substantial computational resources to incorporate two highly relevant models into our evaluation: **AlphaFold3** and **AFsample2** (which is based on AlphaFold2).
>
> The process of generating conformations is computationally intensive. Due to time constraints, we have so far completed the generation of 3,000 conformations for **14 protein targets using AFsample2**. The two additional models were evaluated on the protein conformation landscape diversity of these proteins, incorporating the newly introduced metrics mentioned in the first question. Consequently, our current comparative analysis is limited to these 14 protein targets.
>
> The table below presents a comparative analysis of the **protein conformation landscape diversity** achieved by the different methods.
>
> |**Method** |Energy|**Interaction**|($\\uparrow$)|   |Energy|**Coverage**| ($\\uparrow$)|   |Energy|**Jaccard**|($\\uparrow$)|
> |------------|:---:|:---:|:---:|:---:|:---:|:---:|:---:|:---:|:---:|:---:|:---:|
> | | **5** | **10** | **20** | |**5** | **10** | **20** || **5** | **10** | **20** |
> | AlphaFlow$_{\\mathrm{MD}}^{\\mathrm{Dis}}$ | 2.83 | 59.39 | 108.06 || 0.029 | 0.130 | 0.178 || 0.005 | 0.039 | 0.072 |
> | AlphaFlow$_{\\mathrm{PDB}}^{\\mathrm{Dis}}$ | 6.44 | 64.17 | 115.28 || 0.043 | 0.136 | 0.179 || 0.012 | 0.047 | 0.086 |
> | ESMFlow$_{\\mathrm{MD}}^{\\mathrm{Dis}}$ | 2.33 | 76.33 | 143.72 || 0.022 | 0.143 | 0.237 || 0.004 | 0.043 | 0.089 |
> | ESMFlow$_{\\mathrm{PDB}}^{\\mathrm{Dis}}$ | 6.44 | 64.17 | 115.28 || 0.043 | 0.136 | 0.179 || 0.012 | 0.046 | 0.086 |
> | AlphaFold3 | 7.47 | 60.00 | 104.59 || 0.039 | 0.081 | 0.126 || 0.021 | 0.045 | 0.076 |
> | AFsample2 | 1.14 | 13.86 | 90.43 || 0.004 | 0.047 | 0.116 || 0.003 | 0.020 | 0.070 |
> | BioEmu | **13.00** | 91.11 | 146.78 || 0.079 | 0.178 | 0.250 || **0.031** | 0.089 | 0.138 |
> | ESMDiff | 10.12 | **147.88** | **225.29** || **0.148** | **0.316** | **0.336** || 0.023 | **0.107** | **0.166** |
>
> Our experimental results indicate that the diffusion-based AlphaFold3 achieves performance comparable to AlphaFlow, which employs flow-matching techniques for conformation generation. AlphaFlow, a fine-tuned variant of AlphaFold2, introduces stochasticity through noise-based initialization, enabling the generation of diverse conformations. Given the high similarity between the training paradigms and generative principles underlying AlphaFold3 and AlphaFlow, their performance metrics naturally converge to a comparable level. In contrast, AFsample2, which generates conformations by randomly masking columns in the MSA, demonstrates the weakest performance among the evaluated methods. This outcome suggests that perturbing the MSA in this manner only marginally alters the encoded evolutionary relationships across sequence positions, thereby limiting the exploration of the global conformational landscape. Consequently, AFsample2 predominantly samples local structural variations near the reference fold, restricting its ability to capture broader conformational diversity.
>
> In addition to the newly proposed evaluation metric outlined in our response to the first identified weakness, we have further extended our analysis of physical plausibility by incorporating  **Ramachandran outlier rates**  as an additional assessment criterion. The results, summarized below, underscore the superior structural quality achieved by the newly introduced baselines and reveal a notable trade-off in performance across the different conformation generation methods.
>
> | Model  | Avg. JS-PwD (↓) | Avg. JS-Rg (↓) | Rama Outlier % (↓) |
> | :--------------------------| :--------------:| :-------------:| :-----------------:|
> | **AlphaFold3**           | 0.632           | 0.293          | **2.66%**          |
> | AFsample2                   | 0.490           | 0.311          | 6.61%              |
> | **BioEmu**                    | 0.235           | 0.206          | 10.69%             |
> | **ESMDiff**                 | **0.128**       | **0.104**      | 20.41%             |
> | AlphaFlow$_{\\mathrm{MD}}^{\\mathrm{Dis}}$             | 0.424           | 0.310          | 13.43%             |
> | AlphaFlow$_{\\mathrm{PDB}}^{\\mathrm{Dis}}$            | 0.445           | 0.412          | 9.93%              |
> | ESMFlow$_{\\mathrm{MD}}^{\\mathrm{Dis}}$             | 0.283           | 0.507          | 13.36%             |
> |  ESMFlow$_{\\mathrm{PDB}}^{\\mathrm{Dis}}$             | 0.417           | 0.697          | 12.26%             |
>
> *(Note: Averages are calculated over the same 14 protein targets for which AFsample2 results were available, ensuring a fair comparison across all models.)*
>
> These results yield several key insights that we will now discuss in the revised manuscript:
>
> * **AlphaFold3 and AFsample2 demonstrate superior backbone stereochemistry**, achieving the lowest Ramachandran outlier rates by a significant margin.
>
> * We observe a clear trade-off between geometric similarity and physical plausibility. **ESMDiff**, which is the top performer on distributional similarity (lowest average JS-PwD and JS-Rg), exhibits the highest Ramachandran outlier rate.
>
> * This finding, which is consistent with our original PCPS metric, underscores the value of our multi-faceted evaluation framework in revealing the distinct strengths and weaknesses of each generative approach.
>
> ---
>
> We greatly value your insightful feedback. If you have any additional questions, we will remain fully available to provide detailed clarification.
>
> ---
>
> [1] Lu J, et al. Structure language models for protein conformation generation. arXiv preprint arXiv:2410.18403. 2024.

---

> > ### Author Response · Authors · 2025-08-05
> >
> > Dear Reviewer NMAr,
> >
> > We hope this message finds you well.
> >
> > We are writing to gently follow up on the author rebuttal we submitted last week for Paper #1323. We greatly appreciate the effort that went into your initial reviews, and we have done our best to address your comments and concerns in our response.
> >
> > We understand you are incredibly busy, especially during this phase. We would be grateful if you have a chance to look over our clarifications as you finalize your recommendations.
> >
> > Thank you for your valuable time and effort.
> >
> > Sincerely,
> >
> > The Authors of Paper #1323

---

> > ### Comment · Reviewer_NMAr · 2025-08-06
> >
> > The author's feedback basically answered my question, so I will keep my score.

---

> > > ### Author Response · Authors · 2025-08-07
> > >
> > > Dear Reviewer NMAr,
> > >
> > > Thanks for your reply.
> > >
> > > We appreciate you taking the time to provide the feedback.
> > >
> > > Best regards,
> > >
> > > The Authors of Paper #1323

---

### Official Review · Reviewer_HvTR · 2025-07-10

**Rating:** 4
**Confidence:** 3

**Summary:**

In this paper, the authors try to address a common shortcoming of protein structure prediction methods where most current protein structure prediction methods focus on a single native structure, while proteins naturally sample a diverse ensemble of conformations from the conformational landscape. Existing benchmarks and metrics primarily test static prediction accuracy and don’t systematically assess a model’s ability to generate physically plausible, diverse conformers. To systematically and quantitatively evaluate these methods, the authors present ProteinConformers, a benchmark dataset of 387000 physically refined protein conformations for selected (87) CASP targets. These were generated by taking ~40000 structural decoys from multiple prediction methods and refining them with short all-atom MD simulations (totaling 6 million CPU hours). They evaluate 6 different protein generative models on their dataset using a novel evaluation scheme that evaluates both conformational diversity and physical plausibility. Across their evaluations, they identify that current models work well for interatomic distance prediction but fails on torsion angle sampling.

**Dataset Code Accessibility:**

Partly

**Dataset Code Comments:**

The dataset and code are shared in Hugging Face (https://huggingface.co/datasets/Jim990908/ProteinConformers/tree/main) as zip files causing reduced ease of access. The README document is uninformative and dataset card has no details. The overall accessibility of data and code can be improved for ease of access and use.

**Ethical Considerations:**

No, there are no or only very minor ethics concerns

**Limitations Weaknesses:**

Dataset coverage bias due to decoys come mostly from CASP model predictions. Lacks physics-based coarse-grained sampling, or enhanced sampling methods like metadynamics - landscape coverage is not exhaustive. Generalizability is not rigorously validated.

**Strengths Contributions:**

1. First systematic benchmarking dataset to my knowledge for multi-conformational protein structure evaluation. The metrics for conformational diversity and physical plausibility are useful.
2. Massive dataset of physically plausible conformations spanning native and non-native regions

---

> ### Author Rebuttal · Authors · 2025-07-31
>
> **Dear Reviewer HvTR,**
>
> Many thanks for your valuable comments and questions. We are particularly grateful for the recognition of our contributions, including the introduction of the first systematic benchmarking dataset for multi-conformational protein structure evaluation, the novel metrics for assessing both conformational diversity and physical plausibility, and the creation of a large-scale, physically refined dataset that spans both native and non-native regions. These points highlight the core motivations and strengths of our work. We also appreciate the reviewer’s comments on the current limitations, in response, we have carefully revised the paper and provide detailed point-by-point replies to each of the comments below. These insights have been invaluable in helping us improve the clarity, rigor, and scope of our manuscript.
>
> ---
>
> **[W1] (Limitations Weaknesses)** *Dataset coverage bias due to decoys come mostly from CASP model predictions. Lacks physics-based coarse-grained sampling, or enhanced sampling methods like metadynamics - landscape coverage is not exhaustive. Generalizability is not rigorously validated.*
>
> **[A1]** We sincerely thank the reviewer for this thoughtful comment. We fully agree that comprehensive landscape coverage remains a fundamental challenge in conformational modeling. As the reviewer correctly points out, relying solely on CASP-derived decoys may bias the ensemble toward structure prediction priors. To address this, we incorporated physics-based 3DRobot-generated conformers to increase coverage diversity. We also stated “The decoy set does not incorporate much fragment assembly or physics-based methods to improve diversity” in the section Discussion. We also note that, given the astronomical size of protein conformational spaces, it is theoretically intractable to exhaustively sample all possible states at scale. This limitation is shared across many studies in this field. The suggestion to include coarse-grained or enhanced sampling methods (e.g., metadynamics) is highly valuable. These techniques could enable broader exploration of high-energy or kinetically inaccessible regions at reduced computational cost. However, we found that applying such methods at scale requires carefully selecting physically meaningful collective variables or coarse-grained simulation scheme that generalize across diverse proteins, otherwise, it risks injecting noise into the dataset. Designing such robust protocols is non-trivial and is indeed a key direction for future work. To rigorously validate the generalizability, in future work, we plan to incorporate orthogonal data resources such as NMR ensembles.
>
> At this stage, it is challenging to comprehensively expand the dataset. However, we have made significant improvements to its accessibility and usability. Specifically, we have developed an interactive website that supports searching, filtering, querying, downloading, and visualization of the conformers. We have also elaborated on this in our response to your Dataset Code Comments section. Incorporating the reviewer’s suggestions, we have revised the Discussion section of the manuscript to clear limitations and future directions. We greatly appreciate these insightful suggestions, which help sharpen the roadmap for extending the dataset and rigorously validating its generalizability in future work.
>
> **[W2] (Dataset Code Comments)** *The dataset and code are shared in Hugging Face as zip files causing reduced ease of access. The README document is uninformative and dataset card has no details. The overall accessibility of data and code can be improved for ease of access and use.*
>
> **[A2]** Thank you for the constructive comment. We fully agree with the reviewer that improving ease of access is a critical concern. While the Hugging Face repository primarily serves as a static file hosting platform providing a representative subset of the data (87 ZIP archives), we recognize the limitations this imposes on usability. To address this, we have developed a comprehensive interactive, data-driven web platform that hosts the full ProteinConformers dataset. Per NeurIPS rebuttal policy, we are not allowed to share the URL here, but we have included it in the revised manuscript. This platform significantly enhances data accessibility and user experience. Users can filter conformers based on structural metrics such as TM-score and RMSD, making it easy to retrieve and download specific conformations of interest. The platform also provides interactive 3D visualization of conformers, including alignment views with the native structure to better assess conformational similarity. In addition, we annotate each protein with functional information, to provide more contextual insight into the dataset. These features collectively make the platform far more powerful and user-friendly than static file hosting.
>
> Furthermore, we have prepared an updated version of the README in our Hugging Face repository with clearer documentation and more detailed usage instructions. However, in accordance with NeurIPS rebuttal guidelines, which prohibit including links or making updates to submitted data or code repositories during the review period, we will defer publishing this updated README until after the rebuttal process concludes. We sincerely thank the reviewer for highlighting this important point, which has led to a meaningful and concrete improvement in the accessibility and usability of our dataset. We sincerely thank the reviewer for highlighting this important point, which has led to a meaningful and concrete improvement in the accessibility and usability of our dataset.
>
> ---
>
> **We sincerely appreciate your constructive feedback, which has helped us improve the quality of our work. We hope that our responses have addressed your concerns. Please feel free to reach out with any additional questions during the rebuttal period—we will remain actively available for clarification.**

---

> > ### Author Response · Authors · 2025-08-05
> >
> > Dear Reviewer HvTR,
> >
> > Thank you again for your time and your insightful comments.
> >
> > We submitted our author response last week and hope it helps to clarify the questions you raised. We would be very grateful if you could take our response into consideration during the discussion period.
> >
> > Thank you for your valuable time and effort.
> >
> > Best regards,
> > The Authors of Paper #1323

---

### Note · Authors · 2025-08-12

Dear Reviewers, ACs, SACs, and PCs,

We sincerely thank you for your time and constructive feedback. The review and rebuttal process has substantially strengthened our work. Below, we summarize the key contributions recognized by the reviewers, the clarifications and additional experiments provided during the rebuttal, and our planned revisions for the final version.

**Positive feedback from reviewers:**
- **Novelty & Significance (HvTR, NMAr, owsH, gtWq):** Recognized as the first systematic benchmarking dataset for multi-conformational protein structure evaluation, with novel metrics for diversity and plausibility.
- **Dataset contribution (HvTR, NMAr, owsH, gtWq):** Praised for large-scale, physically refined conformations covering native and non-native regions, filling a key gap in protein conformer benchmarks.
- **Benchmark contribution (HvTR, NMAr, owsH, gtWq):** Valued for introducing new evaluation metrics and providing comprehensive insights through benchmarking six recent models.
- **Clarity & presentation (NMAr):** Commended for clear organization and thorough technical details.

**Key improvements during rebuttal:**
- **Clarifying novelty (HvTR, owsH, gtWq):** Emphasized scope and uniqueness, including broader landscape coverage and energetic annotation.
- **Additional benchmark (NMAr, owsH, gtWq):** Incorporated AlphaFold3 and AFsample2 into comparative analysis, providing new diversity and plausibility results.
- **Evaluation extension (NMAr, gtWq):** Added Jensen–Shannon divergence metrics (JS-PwD, JS-Rg) and Ramachandran outlier rates for stereochemical realism.
- **Discussion clarifications (HvTR, owsH, gtWq):** Expanded discussion of MD coverage limits, Boltzmann distribution assumptions, and future enhanced sampling strategies (e.g., metadynamics).

**Planned revisions for the final version:**
- **Additional results:** Include the AlphaFold3/AFsample2 evaluation, as well as JS-PwD, JS-Rg, and Ramachandran outlier rates metrics results.
- **Expanded discussion details:** Add actionable torsion-angle modeling strategies and broader landscape exploration methods.
- **Easier dataset access:** Launch interactive website with search/filter/download/visualization, improved README, and released code on GitHub.

We appreciate the constructive engagement from all reviewers and ACs. We believe the enhanced analyses, broader benchmarks, and improved accessibility will further increase the impact and usability of ProteinConformers for the community.

---

### Decision · Program_Chairs · 2025-09-18

**Decision:**

Accept (poster)

**Comment:**

This paper presents a large-scale protein conformation benchmark to evaluate the diversity and physical plausibility of AI-generated structures. Protein conformation sampling is an important problem for protein understanding. Existing benchmarks mainly focus on evaluating the static protein structures. All reviewers agree this benchmark is an important contribution for the field of protein modeling and understanding.